# Immunotherapy for Hepatocellular Carcinoma in the Setting of Liver Transplantation: A Review

**DOI:** 10.3390/ijms24032358

**Published:** 2023-01-25

**Authors:** Zurabi Lominadze, Kareen Hill, Mohammed Rifat Shaik, Justin P. Canakis, Mohammad Bourmaf, Cyrus Adams-Mardi, Ameer Abutaleb, Lopa Mishra, Kirti Shetty

**Affiliations:** 1Division of Gastroenterology & Hepatology, Department of Medicine, University of Maryland School of Medicine, Baltimore, MD 21201, USA; 2Department of Medicine, University of Maryland Medical Center, Baltimore, MD 21201, USA; 3Department of Medicine, University of Maryland Medical Center Midtown Campus, Baltimore, MD 21201, USA; 4Department of Medicine, George Washington University School of Medicine and Health Sciences, Washington, DC 20037, USA; 5Department of Surgery, George Washington University School of Medicine and Health Sciences, Washington, DC 20037, USA; 6Cold Spring Harbor Laboratory, Feinstein Institutes for Medical Research, Division of Gastroenterology and Hepatology, Northwell Health, Manhasset, NY 11030, USA

**Keywords:** hepatocellular carcinoma, immunotherapy, liver transplantation, allograft rejection

## Abstract

The emerging field of immuno-oncology has brought exciting developments in the treatment of hepatocellular carcinoma (HCC). It has also raised urgent questions about the role of immunotherapy in the setting of liver transplantation, both before and after transplant. A growing body of evidence points to the safety and efficacy of immunotherapeutic agents as potential adjuncts for successful down-staging of advanced HCCs to allow successful transplant in carefully selected patients. For patients with recurrent HCC post-transplant, immunotherapy has a limited, yet growing role. In this review, we describe optimal regimens in the setting of liver transplantation.

## 1. Introduction

Globally, hepatocellular carcinoma (HCC) is the 3rd leading cause of cancer-related deaths [1]. In the United States, it is the 6th leading and the most rapidly rising cause of cancer deaths [2,3]. Worldwide, the most common etiologies of liver disease predisposing to HCC are chronic viral hepatitis B and C, alcohol-related liver disease, and non-alcoholic or metabolic liver disease [4]. The American Association for the Study of Liver Diseases (AASLD), the European Association for the Study of the Liver (EASL), and the Asian Pacific Association for the Study of the Liver (APASL) all recommend screening for HCC with abdominal ultrasound (with or without alpha fetoprotein (AFP)) in appropriately selected patients: those with cirrhosis of any etiology, and in certain sub-groups with chronic hepatitis B [5,6,7]. However, the sensitivity of AFP as a screening test for detecting HCC is only about 41–65% (based on a systematic review of patients with chronic hepatitis C) [8], and the sensitivity of ultrasound for detecting early-stage HCC is approximately 63% [9]. In addition to the limitations in sensitivity of existing screening tests, adherence to the screening guidelines themselves is sub-optimal: a systematic review and meta-analysis of studies from around the world found a 24% pooled proportion of patients undergoing surveillance [10]. 

Given these challenges in detecting early-stage HCC, it is no surprise that the prognosis of HCC varies widely. Patients with resectable lesions have a 5-year survival of approximately 70% following surgical intervention [6,11,12], whereas the median overall survival of untreated terminal-stage HCC (Barcelona Clinic Liver Cancer stage D) is only 1.6–6 months [2]. Patients between these two extremes comprise a heterogeneous group, with multiple possible treatment pathways. These include thermal ablation techniques (microwave and radiofrequency), bland embolization, trans-arterial chemoembolization (TACE), trans-arterial radioembolization (TARE), external beam radiation therapy (EBRT) modalities including proton beam therapy, liver transplantation (LT), and systemic treatments. For patients who qualify for LT, whether upon initial diagnosis based on the Milan criteria, or after “down-staging” with local-regional therapies, 5-year survival is excellent at 70–80% [4,13,14].

The emerging field of immunotherapy has brought exciting developments in the treatment of numerous cancers and has the potential to revolutionize the approach to intermediate and advanced-stage HCC. The landmark Checkmate 040 trial led to the Food and Drug Administration (FDA) approval for the use of the immune checkpoint inhibitors (ICIs) nivolumab (directed against programmed death-1, or PD-1) plus ipilimumab (directed against cytotoxic T-lymphocyte-associated-protein 4, or CTLA-4) for the treatment of advanced HCC previously treated with sorafenib [15]. Shortly thereafter, the IMbrave150 clinical trial led to the FDA approval of atezolizumab (a programmed death-ligand 1, or PD-L1 inhibitor) plus bevacizumab (a monoclonal antibody targeted at vascular endothelial growth factor, or VEGF) as first line therapy for advanced stage HCC [16]. Recently presented phase 3 data (HIMALAYA trial) with the novel immunotherapeutic combination of durvalumab (another PD-L1 inhibitor) plus tremelimumab (another anti-CTLA-4 antibody) demonstrated better overall and progression-free survival compared to sorafenib as first line treatment [17,18]. The FDA approved this combination on 21 October 2022, making it an attractive first-line option in advanced HCC.

The current choice of therapy for HCC is dictated by hepatic synthetic function, presence or absence of portal hypertension, patient performance status, and tumor burden. The American Society of Clinical Oncology guidelines recommend atezolizumab-bevacizumab or the tyrosine kinase inhibitors (TKIs) sorafenib or lenvatinib as first-line therapy [19]. The choice of second-line therapy is dependent on initial treatment. For patients who progress on TKIs, second-line therapy options include another TKI (such as cabozantinib or regorafenib), immunotherapy (atezolizumab-bevacizumab, pembrolizumab, or nivolumab), or ramucirumab (a VEGF receptor inhibitor) for those with AFP levels greater than 400 ng/mL. Those treated with durvalumab-tremelimumab or atezolizumab-bevacizumab as first-line therapy may be offered TKIs such as sorafenib, lenvatinib, regorafenib or cabozantinib [19].

Multiple studies have established the benefit of immunotherapeutic agents in the treatment of HCC, and many studies are ongoing [20,21]. Preliminary studies of chimeric antigen receptor (CAR) T cells have also shown promise in treating advanced or recurrent HCC [22,23]. Finally, combining immune-oncology (IO) and interventional radiology (IR) treatments in a complementary fashion has led to the proposition for a more formalized “IR-IO” approach in treating early and intermediate stage HCC [24].

However, a controversial aspect of using immunotherapy against HCC arises in the setting of LT: specifically, what is the role of immunotherapy as an adjunct to local-regional therapies to down-stage HCC to serve as a bridge to LT? And, what role, if any, should immunotherapy have for treating recurrent HCC following LT? In this review, we will focus on the intersection of immunotherapy and LT, and the increasing role these new treatments may play both before and after transplant.

## 2. Immune-Focused Pathophysiology of HCC

HCC usually occurs in the setting of advanced chronic liver disease, such as advanced hepatic fibrosis or cirrhosis, or chronic hepatitis B with necro-inflammatory activity. In these settings, a plethora of inflammatory and immune mediated responses contribute to the development and survival of HCC. 

Kupffer cells, which form part of the reticuloendothelial system of the liver, are the resident macrophages found in hepatic sinusoids. Pattern recognition receptors found on these cells are stimulated by pathogen-associated molecular patterns (PAMPs) leading to the expression of pro-inflammatory cytokines such as IL-6, as well as immunomodulatory molecules including IL-10 and TGF-β [25]. Mouse studies have shown liver sinusoidal endothelial cells expressing low levels of major histocompatibility complex (MHC) class II molecules CD80 and CD86. Vital for helper T cell differentiation, the lack of these molecules provides a mechanism of immune escape for HCC [25,26].

Another mechanism of immune evasion by cancer cells is through the upregulation of T regulatory cells (Tregs) and myeloid-derived suppressor cells (MDSCs). CD4+CD25+ Tregs account for 5–10% of the CD4+ T cell population in the human body. These cells are known to suppress both CD4+ and CD8+ T cells through various means including secretion of TGF-β, IL-10, and IL-35 [27,28,29]. CTLA-4, expressed and activated on Tregs, has a higher affinity for CD80 and CD86 which acts as a competitive inhibitor to CD28, a T cell activator. Through this binding, CTLA-4 can activate indoleamine-2,3-dioxygenase (IDO) and subsequently suppress T cell mediated cytotoxic immune responses [30]. A meta-analysis from 2014 including twenty-three studies showed patients with HCC had 87% more circulating Tregs compared to healthy controls [29].

Similarly, MDSCs have been shown to suppress CD4+ and CD8+ T cells, inhibit natural killer cell cytotoxicity, and prompt the development of Tregs [31]. Accumulation of MDSCs in the liver leads to interaction with Kupffer cells, inducing PD-L1 expression on their surface. PD-L1 interacts with PD-1 and leads to suppression of IFN-γ [31]. MDSCs along with tumor associated macrophages (TAMs) also play an important role in angiogenesis: secretion of VEGF, platelet-derived growth factor (PDGF), and matrix metalloproteinase-9 (MMP-9) by TAMs and MDSCs leads to increased tumor vasculature, providing nutrients for tumor growth [32,33]. In turn, increased expression of PD-1 on T cells has been associated with incomplete response to HCC local-regional therapy and impaired survival, possibly mediated by the phenomenon of T cell exhaustion [34].

Through secretion by Tregs and MDSCs, anti-inflammatory cytokines promote tumor growth and are associated with a worse prognosis [35]. Some of the cytokines studied in HCC include the previously mentioned IL-10 and TGF-β. While IL-10 has been shown to be increased in patients with HCC compared to those with only cirrhosis or healthy controls, it is also elevated in those with chronic viral hepatitis [36]. Therefore, this immune-suppressive cytokine may be useful as an HCC tumor marker, though with limited utility in those with chronic viral hepatitis. Meanwhile, TGF-β, initially thought to be tumor suppressive, functions as a tumor promoter in later stages of HCC. Produced by tumor cells, macrophages, or Tregs, it downregulates the antitumor response by various mechanisms: it inhibits the activation of dendritic cells, promotes M2 polarization of TAMs, impairs the effector functions of T cells and NK cells, and promotes the generation of induced Treg cells [37]. One recent mouse study demonstrated that FGFR4 contributes to tumor proliferation and invasion through activation by TGF-β1 in the extracellular signal-related kinase (ERK) pathway; silencing FGFR4 expression was found to inhibit this activity [38]. Higher levels of TGF-β1 have been shown to be correlated with higher tumor grade, shorter survival, and overall poor prognosis [39,40]. In addition, higher TGF-β1 and TGF-β levels, respectively, have been associated with poor response to the TKI sorafenib and ICI pembrolizumab [41,42].

Several signaling pathways have also been studied in the development and progression of HCC. These include the Ras/Raf/MAPK, PI3/AKT/mTOR, JAK/STAT, and the Ubiquitin-Proteasome pathway. Transcription factors including c-myc and c-jun, among others, are associated with these pathways and are involved in 30–60% of HCCs [43]. Of these, one of the most crucial is the Wnt/β-Catenin pathway, with 20–40% of HCC cases harboring mutations within this complex [43]. Specifically, mutations in CTNNB1 which encodes β-catenin, as well as AXIN1 and APC encoding other important components, have been shown to occur in up to a combined 35% of human HCC samples [44]. Furthermore, activation of this pathway is shown to mediate resistance to ICI treatment for HCC, with upregulation of this signaling pathway correlating to lower overall survival rates and resistance to anti-PD1 therapies [45,46]. 

Directly targeting immune evasion by blockade of PD-1, PD-L1, and CTLA-4, has led to the development of various ICIs, which have proven to be successful immunotherapies for multiple cancers, including HCC. These will be discussed in more detail in the following section.

## 3. Mechanisms of Action of Immunotherapy for HCC

As described above, tumor cells often utilize multiple resistance mechanisms to evade the host immune system. PD-L1 is a protein that can be expressed on tumor cells that contributes to T cell exhaustion, especially in the tumor microenvironment [15]. Normally, PD-L1 on antigen presenting cells (APCs) binds to PD-1, also known as CD279, which is an inhibitory receptor expressed on activated T cells. This mechanism helps establish both central and peripheral immune tolerance. When bound, PD-1 activation results in the suppression of T cell mediated immune recognition. Thus, this mechanism can be harnessed by cancer cells to achieve immune evasion. Nivolumab, camrelizumab, tislelizumab, and toripalimab are all monoclonal IgG4 antibodies that bind PD-1 and prevents its interaction with PD-L1 [47,48,49,50]. Similarly, durvalumab is a monoclonal IgG1 antibody that binds PD-L1 and prevents its interaction with PD-1 [51]. As such, all the aforementioned agents halt the inactivation of immune cells by tumor cells, allowing for the restoration of an immune attack on malignant tissues.

Similarly, CTLA-4 inhibits activated T cells in lymphoid organs via two mechanisms. First, CTLA-4 presentation is upregulated on T cells in response to their activation. CTLA-4 competes with CD28 for binding B7 ligand subtypes CD80 (B7-1) and CD86 (B7-2) presented on APCs. Without CD28 binding, T cells are not able to undergo activation via co-stimulation. Additionally, binding of Treg associated CTLA-4 to APCs results in suppression of T cell activity and is a potent negative regulator of T cell mediated antitumor immune responses [52]. By countering these mechanisms, novel anti-CTLA4 monoclonal antibodies such as ipilimumab and tremelimumab result in increased T cell antitumor activity [52,53].

The Checkmate 040 trial led to FDA approval for the use of a combination of ICIs nivolumab plus ipilimumab for the treatment of unresectable HCC previously treated with sorafenib [15]. These agents promote immune responses against tumor cells in a complementary manner by reducing inhibitory signals on T cells. In contrast, sorafenib is a multikinase inhibitor that serves to inhibit Raf, VEGF receptor (VEGFR), and PDGF receptor (PDGFR) [54]. Raf, when activated by Ras, has a variety of tumor-promoting cellular effects. VEGF is a circulating protein whose expression is upregulated under hypoxic physiological conditions. However, tumor expression of VEGF receptor and its coreceptors aids in vascularization of malignant tissue. VEGF, when bound to VEGFR, results in increased endothelial cell mitosis, increased migration, inhibition of apoptosis, and reversal of endothelial cell aging. PDGFR, when bound to PDGF, has several effects on endothelial cell apoptosis and migration in both normal and tumoral conditions [55].

The IMbrave150 clinical trial led to the FDA approval of atezolizumab plus bevacizumab as first line therapy for advanced stage HCC over sorafenib [16]. Unlike nivolumab, atezolizumab directly targets PD-L1 to prevent it from interacting with PD-1 and CD80, resulting in reversal of T cell suppression [16]. In contrast, bevacizumab is a monoclonal antibody that selectively binds to VEGF and inhibits its interaction with cell surface receptors. Inhibition of this pathway limits blood supply to tumor tissue and may aid in the delivery of other chemotherapeutic medications [56].

(CAR) T cell therapy has also shown some promise in treating advanced or recurrent HCC [22,23]. These therapies involve the modification of T cells to target surface moieties on HCC cells. Glypican-3 (GPC3) is a surface protein highly expressed in HCC but with low expression in other tissues, making it a potential target for CAR T cells [23]. Though these cells have been shown to be a safe and effective treatment for HCC, this treatment modality is subject to interference from ongoing chemotherapy or immunosuppression. However, there are potential methods to create T cells that are resistant to ongoing immunosuppressive treatment [22].

Other promising immune strategies studied in HCC include adoptive T-cell transfer and vaccine therapy [57,58]. However, these approaches have not yet demonstrated consistent clinical activity against HCC and have not been studied in the liver transplant setting.

## 4. Safety of and Role of Immunotherapy for HCC Pre-LT

Immunotherapy has shown promising objective response rates in clinical trials, with a complete response rate of 5.5% [59]. These findings have led to FDA approval of multiple immunomodulating drugs for the treatment of progressive HCC both as first line agents and for refractory disease since 2017 [60]. A unique situation arises when HCC patients that are not clearly within the LT pathway, but are eligible for immunotherapy, have a favorable response to treatment and eventually meet transplant criteria. This so-called “accidental neoadjuvant” immunotherapy was often given as destination treatment, but when certain patients had a dramatic clinical response, they were eventually listed for transplant [61]. There is a growing list of case reports showing positive outcomes from transplantation post-immunotherapy, but there is a clear lack of data from large scale clinical trials showing superior outcomes in patients who have had immunotherapy prior to transplantation over standard bridging therapy plus transplantation [60]. ICIs dominate the field of immunotherapy while adoptive cell transfer (ACT) and vaccination therapy are rarely described in the literature [59].

As previously described, the immune checkpoint pathways CTLA4—CD80/CD86 and PD-1—PD-L1 inhibit T cell activation thus maintaining peripheral tolerance and helping cancer cells evade cytotoxic T cell mediated death. ICIs potentiate anti-tumor activity through prevention of the interaction between these receptors and their ligands, creating a “hyperactive” immune state leading to activation of T cells and subsequent killing of cancer cells [60,62]. In patients who will not receive a liver transplant, modulating the immune system in this way has the positive results of stabilizing or decreasing tumor burden. For patients who become eligible for transplant after immunotherapy, it is unclear how the modulated immune system will react to an allograft in the setting of immunosuppression. When immunotherapy is held during the peri-transplant period there is thought to be a “cooling off” period wherein the immune system can return to its normal state. After transplantation, immunosuppression is initiated, suppressing the effects of ICIs [60].

Pembrolizumab, nivolumab, toripalimab, durvalumab, camrelizumab, and ipilimumab have all been used as a bridging therapy prior to LT. Combination therapy with atezolizumab and bevacizumab is proposed as a first-line systemic therapy for HCC, according to a recently published guideline [63]. The specific characteristics of each patient and tumor should play a major part in determining the best treatment regimen [59]. While combination therapy consisting of TKI + ICI for downstaging of HCC to transplant has not been studied, the neoadjuvant use of cabozantinib + nivolumab did successfully allow 12 of 15 patients in one study to achieve margin-negative surgical resection of locally-advanced HCC [64]. Although these patients were not transplant candidates, one could speculate on the potential utility of such a regimen in transplant candidates, and upcoming studies will answer this question. 

### 4.1. Efficacy

The existing data, although limited to case reports, has shown bridging immunotherapy to have remarkable efficacy in terms of down-staging patients into Milan criteria, thus making them eligible for transplant. However, its safety in the immediate pre-transplant period is still under debate, due to several instances of severe rejection leading to graft loss and patient death following bridging therapy with ICIs [65,66]. The efficacy of PD-L1 receptor blockers usually appears within 3 months of initiation and lasts for a prolonged period even after immunotherapy is withdrawn [67]. The prolonged effect is due to both an extended half-life of the drugs, shown to be up to 27 days, as well as the prolonged duration of T cell activation. Sustained occupancy of over 70% of PD-1 molecules on circulating T cells has been seen for up to 2 months following a single infusion of nivolumab [68]. Table 1 summarizes published case studies of patients that were treated with ICIs prior to LT. Qiao et al. describe seven patients who received neoadjuvant pembrolizumab or camrelizumab in combination with lenvatinib therapy prior to successful liver transplantation [62]. Although one patient experienced graft rejection after LT, his liver function recovered with adjustment of his immunosuppression regimen. Tabrizian et al. describe a single center trial involving nine patients with recurrent HCC following liver resection as a primary treatment. These patients were given nivolumab and successfully bridged to liver transplant, with one third of the tumors demonstrating nearly full regression on explant histology [69]. Schwacha-Eipper et al. report a patient with recurrent HCC that had progressed to systemic therapy with sorafenib who underwent a successful liver transplant following 34 cycles of nivolumab [70]. There was no evidence of allograft rejection or recurrence of the tumor at the one-year mark following transplant. Abdelrahim et al. describe a patient treated with atezolizumab and bevacizumab combination therapy for poorly differentiated HCC preceding a successful LT [71]. The Mayo Clinic in Arizona reported on the successful utilization of nivolumab in a patient with HCC secondary to alcohol-related cirrhosis who developed new lesions despite Y-90 radioembolization. He responded well with a decline in AFP, no newly formed lesions and underwent a successful rejection-free transplant 9 weeks later. The explant pathology revealed no evidence of residual HCC, and the patient remained cancer free with no evidence of rejection at 12 months [72]. Five cases of successful nivolumab treatment for patients with T2 and T3 HCC are referenced in the case series from the University of California, San Diego [73]. The length of time spent withdrawing immunotherapy varied from ten days to six months. Four of the five patients were alive at the time of publication, with one requiring retransplant secondary to massive hepatic necrosis.

### 4.2. Side Effects and Adverse Events

The most common adverse effects of immunotherapy include fatigue, diarrhea, rash, pruritus, and decreased appetite [21]. The much-feared side effects of hepatotoxicity and graft rejection are clinically indistinguishable without a liver biopsy [59]. Immunotherapy is thought to induce severe graft rejection in transplant recipients due to the activation of the innate immune response [71,74]. This can be mitigated by an adequate washout period, modalities such as plasmapheresis, as well as increased immunosuppression post-transplant. Nordness et al., Abdelrahim et al., and Aby and Lake describe cases of acute rejection in which ICI therapy was withdrawn 7, 8, and 16 days prior to LT, respectively, necessitating the importance of an adequate washout period [66,71,75]. Given the fact that PD-1 occupancy remains high even after 4–5 half-lives, a shorter washout period should be avoided [59,68,76]. On the other hand, successful transplantations were performed by Tabrizian et al., 1 day and 2 days after the last nivolumab infusion [69]. These successes were thought to possibly be related to the multiple blood transfusions the transplant recipients received in the setting of significant intraoperative blood loss. This suggests the potential utility of modalities such as plasmapheresis in order to accelerate washout [68]. 

**Table 1 ijms-24-02358-t001:** Summary of ICI use pre-LT.

Reference	# of Patients	Drug(S) Used	Lines of Treatment Prior to ICI	Washout Period (Days)	Successful LT at 12 Months?	Rejection?	Tumor Regression/ Tumor Necrosis on Explant?
Qiao [62]	7	pembrolizumab or camrelizumab in combination with lenvatinib	unknown	40 (average)	In 7/7	Yes—in 1/7 (reversed with altered IS)	unknown
Tabrizian [69]	9	nivolumab	0–7	1–253	In 9/9	Yes—in 1/9 (reversed with altered IS)	In 3/9 patients
Schwacha-Eipper [70]	1	nivolumab	unknown	105	yes	No	unknown
Abdelrahim [71]	1	atezolizumab and bevacizumab	unknown	60	yes	No	unknown
Lizaola-Mayo [72]	1	nivolumab	1 (TARE)	unknown	yes	No	unknown
Nordness [66]	1	nivolumab	4 (laparoscopic resection, sorafenib, TARE, TACE)	8	no	Yes—fatal hepatic necrosis, death on POD 10	yes
Schnickel [73]	5	nivolumab	unknown	10–183	In 4/5	Yes—in 1/5 (successful retransplant for massive hepatic necrosis)	unknown
Sogbe [76]	1	durvalumab	unknown	>90	yes	No	unknown
Chen [65]	1	toripalimab	3	93	no	Yes—fatal hepatic necrosis, death on POD 3	unknown
Aby & Lake [75]	1	nivolumab	4 (TARE, TACE, MWA, sorafenib)	16	yes	Yes—treated successfully	yes

IS: immunosuppression; POD: postop day; MWA: microwave ablation.

### 4.3. Patient Selection and Duration of Therapy

In the setting of LT, there are no guidelines or strong consensus on how and when to use ICIs, how to select the most appropriate patients for this therapy, and how to predict those at highest risk of rejection post-LT. Similarly, as there are no comparative or head-to-head trials, it is not possible at this time to determine which specific immunotherapeutic agent (or agents) are best used in this situation. The total duration of therapy and the right time for discontinuation or tapering of immunotherapy is likewise unclear. Consideration of a tapering strategy should be prompted by disease stabilization or regression and timing of listing as a candidate for LT [59]. Given the experience so far, it is reasonable to recommend a washout period measured in weeks to months, rather than days, to reduce the risk of post-transplant graft loss.

### 4.4. Markers of Response

PD-L1 expression in tumor cells may be used as an indicator of response to ICIs. The CheckMate 040 trial, however, showed that tumor response could occur despite the lack of PD-L1 expression [15]. Furthermore, Nordness et al. describe PD-L1 expression as a possible secondary phenomenon following ICI-associated rejection [66]. Tumor infiltrating lymphocyte (TIL) density may predict clinical response to ICIs since responders had a higher density of CD8+ TILs in their tumor microenvironment than non-responders [59]. Also, gut microbiota may have a significant impact on treatment response, and the use of PD-1 inhibitors in conjunction with fecal microbiota transplant may enhance ICI efficacy [59]. The likelihood of a tumor response to ICIs was quantified in the KEYNOTE-224 trial by using a score that incorporated the ratio of PD-L1 positive tumor and immune cells to the total number of viable tumor cells in HCC samples [77]. Absolute lymphocyte count and T cell activation marker inducible co-stimulator are two additional postulated biomarkers that could be used for determining the response to CTLA-4 blockade [59,78].

### 4.5. Current Studies

Table 2 summarizes the current clinical trials investigating immunotherapy as bridging treatment for HCC towards the goal of transplant. There is currently insufficient evidence to propose a specific regimen or recommendation, but as these trials produce results it may become clear which patients should receive immunotherapy prior to transplantation, as well as the optimal length of treatment, and the minimum safe washout period.

## 5. Safety of and Role of Immunotherapy for HCC Post-LT

HCC in LT recipients recurs in about 10–20% of patients, over a median interval of 13–14 months, and with a median survival of about 1 year [14,79,80,81,82]. Known risk factors for tumor recurrence include certain clinical features related to both the patient and pre-transplant treatment, as well as tumor characteristics on explant such as the degree of differentiation and microvascular invasion [83,84]. Immunosuppression is crucial to prevent allograft rejection, but also increases the overall risk of developing malignancy [85].

Post-LT treatment guidelines for recurrent HCC are lacking. Available therapies include surgical resection, ablation, radiation, local and systemic chemotherapies, and immunotherapies [86]. However, the safety and efficacy of immunotherapy in liver transplant recipients is not entirely understood. All the registry trials that led to approval of immune-therapeutic agents for HCC excluded liver transplant recipients. Therefore, most of the data on immunotherapy for HCC after LT is drawn from case reports and case series [85]. 

Immune checkpoints are involved in the immune tolerance required for allograft survival, thus, the addition of ICIs in liver transplant recipients may lead to allograft rejection [87]. Conversely, the use of immunosuppression may mitigate the effect of ICIs [88]. In their systematic review, Kumar et al. found that allograft rejection occurred in 39% of liver transplant recipients following ICI therapy [87]. In addition to graft rejection, patients on immunotherapy are at increased risk of ICI-associated hepatitis and colitis, among other complications [89]. The most commonly used ICIs for HCC post-LT are the PD-1 and CTLA-4 antibodies, with varying safety and efficacy profiles, as described in Table 3.

The anti-PD-1 antibodies that have been used for post-transplant HCC include nivolumab and pembrolizumab. Au and Chok conducted a systematic review exploring many of the aforementioned case reports in Table 3 of LT recipients who received either nivolumab or pembrolizumab for recurrent HCC [102]. They identified 19 patients with recurrent HCC, of whom 14 were treated with nivolumab and five with pembrolizumab. Among these 19 patients, six suffered rejection and one patient suffered early mortality unrelated to rejection. Notably, the overall objective response rate was 11%, the median progression free survival was 2.5 ± 1.0 months, and the median overall survival was 7.3 ± 2.7 months after immunotherapy. When comparing pembrolizumab to nivolumab, pembrolizumab was used as an earlier line of therapy (median second line vs. third line, *p* = 0.03) and was associated with a higher rate of complete response (0% vs. 40%, *p* = 0.03), less progressive disease (50% vs. 20%, *p* = 0.03), better progression free survival (1.3 ± 1.1 vs. 12.4 months, *p* = 0.004), and greater overall survival (4.0 ± 3.4 vs. 19.2 months, *p* = 0.006). Additionally, pembrolizumab was found to have a non-significant trend towards fewer early mortalities (36% vs. 0%, *p* = 0.12) [102]. Interestingly, it was observed that patients further out from LT had a lower risk of rejection when treated with immunotherapy, as compared to patients with recent LT. However, most HCC recurrences occur early after liver transplantation, limiting the applicability of this observation. Notably, this meta-analysis demonstrated that the overall rejection rate was high (32%) and was associated with a dramatically increased rate of organ failure and early mortality (56% in patients with rejection).

There is limited data on the use of CTLA-4 inhibitors in treating HCC post-LT. Pandey and Cohen describe a case of ipilimumab to treat recurrent HCC six years after LT [99]. Upon diagnosis, the patient underwent treatment with sorafenib and NanoKnife ablation. However, the disease progressed, and she was initiated on ipilimumab 7.5 years after OLT while continuing tacrolimus. Her treatment was complicated by a transient episode of grade 2 liver enzyme elevation, but she experienced no other immune-related adverse events and had resolution of disease.

As of the date of this review, there is not enough data on the efficacy and safety of ICIs to treat HCC in the post-LT setting to make definitive recommendations. The combination of immunosuppression and immunotherapy poses a therapeutic challenge given their theoretical antagonistic effects. The current literature demonstrates that ICIs are associated with high rates of graft rejection and a significant risk of mortality. It is interesting to note that increasing time intervals between LT and initiation of ICIs may lower the risk of rejection.

## 6. Conclusions and Future Directions

Although immunotherapies have revolutionized the treatment of advanced HCC, their role in the setting of transplantation is yet to be precisely defined. Prior to transplant, immunotherapies are likely to play an important role in carefully selected patients outside current transplant criteria, with progressive disease despite existing treatment modalities. In such patients, these novel agents may allow patients hitherto deemed ineligible for transplant to undergo successful down-staging. However, the optimal treatment regimens, acceptable treatment durations, as well as safe timing of immunotherapy withdrawal in relation to LT is currently unknown. The optimal approach to immunosuppression management in these patients is also yet to be determined. For those patients with recurrent HCC post-LT, immunotherapy can be considered as a treatment of last resort, given the high rates and severities of rejection. Further research is needed to determine the safest approach to administering immunotherapy post-LT.

In addition to traditional clinical trials, exciting developments in the field of machine learning have allowed the conduct of so-called “virtual clinical trials” using a quantitative systems pharmacology model. Such a virtual clinical trial for ICIs in advanced HCC was recently published, possibly heralding the use of such models to complement traditional research to answer these challenging questions [103].

## Figures and Tables

**Table 2 ijms-24-02358-t002:** Current clinical trials of immunotherapy for HCC pre-LT.

Name	Trial #	Type of Study	# of Participants	Agent(s)	Primary Endpoint(s)	Status
Durvalumab and Tremelimumab for Hepatocellular Carcinoma in Patients Listed for a Liver Transplant	MEDI4736- NCT05027425	Single-arm, open-label, Phase II, multicenter clinical trial	30	Durvalumab & tremelimumab	Rejection rates	Recruiting. Estimated primary completion 12/7/25
Hepatic Arterial Infusion Chemotherapy Combined With Targeted Therapy and Immunotherapy as Down-stage Therapy of Liver Transplant	HLPDTLT- NCT05475613	Prospective single center, open-label, non-randomized, single arm exploratory study	75	Lenvatinib & PD-1 inhibitor	2-year relapse-free survival	Not yet recruiting. Estimated primary completion 7/30/25
Effect of PD-1/PD-L1 Inhibitor Therapy Before Liver Transplantation on Acute Rejection After Liver Transplantation in Patients With Hepatocellular Carcinoma	NCT05411926	Single-center, prospective, non-interventional cohort study based on real world data	30 cohort 30 controls	PD-1/PD-L1 inhibitor monotherapy	Incidence of and severity of ACR after LT. Cellular immune function after LT. Dose and drug concentration of tacrolimus.	Recruiting. Estimated primary completion 3/2023
Sequential TACE and SBRT Followed by ImmunoTherapy for Downstaging HCC for Hepatectomy	START-FIT- NCT03817736	Prospective phase II, single arm clinical study	33	ICI (unspecified)	Number of patients amendable to curative surgical intervention (resection or LT) after down-sizing of tumor(s) by intervention	Active, not recruiting. Estimated primary completion 6/14/22
Pembrolizumab and Lenvatinib in Participants With Hepatocellular Carcinoma (HCC) Before Liver Transplant	PLENTY202001-NCT04425226	Randomized clinical trial	192	Pembrolizumab & lenvatinib	Recurrence-free survival (RFS) up to ~4 years	Recruiting. Estimated primary completion 12/30/22
Atezolizumab and Bevacizumab Pre-Liver Transplantation for Patients With Hepatocellular Carcinoma Beyond Milan Criteria	NCT05185505	Single Group Assignment, Open Label	24	Atezolizumab & bevacizumab	Proportion of patients receiving LT experiencing Acute Rejection—within 1 year after LT	Not yet recruiting. Estimated primary completion 6/17/26
Durvalumab and Lenvatinib in Participants With Locally Advanced and Metastatic Hepatocellular Carcinoma	Dulect2020-1 NCT04443322	Single Group Assignment, Open Label	20	Durvalumab & lenvatinib	1. Progression Free Survival (PFS)—3 year 2. Recurrence-Free Survival (RFS)—4 year	Recruiting. Estimated primary completion 12/31/21
Neoadjuvant Combination Therapy of Lenvatinib and TACE for Transplant-Eligible Large Hepatocellular Carcinoma Patients	NCT05171335	Non-Randomized, Single Group Assignment, Open Label	50	Lenvatinib	Percent tumor necrosis at time of transplant surgery	Enrolling by invitation. Estimated primary completion 6/30/26

ACR: acute cellular rejection; SBRT: stereotactic body radiotherapy.

**Table 3 ijms-24-02358-t003:** ICIs used post-LT.

Study (Year of Publication)	Patient Sex	Patient Age	Years after LT	ICI Used	Prior Sorafenib?	Response	Overall Survival (Months)	Graft Rejection
Friend (2017) [90]	Male	20	4	Nivolumab	Yes	n/a	1	yes
Friend (2017) [90]	Male	14	3	Nivolumab	Yes	n/a	1	yes
DeLeon (2018) [91]	Male	68	1.1	Nivolumab	Yes	n/a	<1	yes
DeLeon (2018) [91]	Male	57	2.7	Nivolumab	Yes	no	1.2 *	no
DeLeon (2018) [91]	Male	56	7.8	Nivolumab	Yes	no	1.1 *	no
DeLeon (2018) [91]	Female	35	3.7	Nivolumab	Yes	no	1.3 *	no
DeLeon (2018) [91]	Male	64	1.2	Nivolumab	Yes	n/a	<1	no
Gomez (2018)—abstract [92]	Male	61	2	Nivolumab	Yes	n/a	n/a	yes
Kumar (2020) [87]	Male	64	2	Nivolumab	Yes	n/a	n/a	yes
De Toni & Gerbes (2017) [93]	Male	41	0.9	Nivolumab	Yes	yes	7	no
Varkaris (2017) [94]	Male	70	8	Pembrolizumab	Yes	no	3	no
Al Jarroudi (2020) [95]	Male	70	2.75	Nivolumab	Yes	no	4	n/a
Al Jarroudi (2020) [95]	Female	62	1	Nivolumab	Yes	yes	n/a	no
Al Jarroudi (2020) [95]	Male	66	5	Nivolumab	Yes	no	n/a	no
Rammohan (2018) [96]	Male	57	4.3	Pembrolizumab	Yes	yes	10 *	no
Gassman (2018) [97]	Female	53	3	Nivolumab	Yes	no	<1	yes
Nasr (2018) [98]	Male	63	4.6	Pembrolizumab	Yes	yes	25 *	no
Pandey & Cohen (2020) [99]	Female	54	7	Ipilimumab	Yes	yes	27 *	no
Anugwom & Leventhal (2020) [100]	Male	62	1	Nivolumab	Yes	no	2	no
Amjad (2020) [101]	Female	62	2	Nivolumab	No	yes	24 *	no

* Alive at time of publication.

## Data Availability

Data sharing not applicable. No new data were created or analyzed in this study. Data sharing is not applicable to this article.

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
