# Peer review of "Immunotherapy for Hepatocellular Carcinoma in the Setting of Liver Transplantation: A Review"

_ijms, 2023, doi:10.3390/ijms24032358_

Round 1

Reviewer 1 Report

The manuscript is fluent and readable in English. The anthors aimed to review the immunotherapies in the setting of liver transplantation (LT). However, immunotherapies except immune checkpoint inhibitors (ICIs), such as adoptive cell transfer and vaccine therapy, were little addressed. Though these immunotherapies are rarely used in pre- and post-LT, there are accumulative related researches in endstage hepatocellular carcinoma (HCC). In the section of Immune-focused pathophysiology of HCC, the authors discussed chemokines, cytokines and signaling pathways which involved in HCC tumorigenesis, they should also clarify why these molecules or genes were chosen for discusssion and their mechanism in immune evasion or roles in immunotherapy. As ICIs, combined with or without monoclonal antibodies targeting vascular endothelial growth factor, are now the first-line systemic regimens, there are more and more downstaging cases for LT after immunotherapy, but cases of immunotherapy for HCC recurrence after LT are still limited. The authors listed those cases, but provided little new contribution in reviewing immunotherapy for HCC pre- and post-LT, as recent system reviews, published in World J Gastrointest Oncol and World J Gastrointest Surg, already described and analyzed these cases comprehensively. 

Author Response

  1. Immunotherapies except immune checkpoint inhibitors (ICIs), such as adoptive cell transfer and vaccine therapy, were little addressed. Though these immunotherapies are rarely used in pre- and post-LT, there are accumulative related research in endstage hepatocellular carcinoma (HCC)

Response: We have inserted statements to reference these immunotherapies, mentioning that they have not been studied in organ transplantation.

  1. In the section of Immune-focused pathophysiology of HCC, the authors discussed chemokines, cytokines and signaling pathways which involved in HCC tumorigenesis, they should also clarify why these molecules or genes were chosen for discussion and their mechanism in immune evasion or roles in immunotherapy.

Response: we have removed one section, and inserted statements outlining the specific effects of the more important signaling pathways in immune evasion and interactions with immunotherapy.

  1. Cases of immunotherapy for HCC recurrence after LT are still limited. The authors listed those cases, but provided little new contribution in reviewing immunotherapy for HCC pre- and post-LT.

Response: We acknowledge that our review offers few fresh insights, but are limited by the current literature in this area. Most studies are in the form of case reports and case series which makes it difficult to extrapolate the data to clinical practice. There is a need for rigorously controlled studies, and we hope that summarizing our current state of understanding of the field will lead to further research in this area.

Reviewer 2 Report

This manuscript summarized the current knowledge of immunotherapy for HCC, including the mechanisms of action; the safety of and role of immunotherapy for HCC pre-LT and post-LT. As immunotherapy is the revolutionary approach and hotspot for advanced HCC treatment, it is important to investigate its role in the setting of LT.

Overall, the manuscript is well prepared. But there are still a few points that should be addressed:

(1)  Page number and line number should be provided.

(2)  In the first and fifth paragraphs of page 3, the authors described that the IL-10 and TGF-b are pro-inflammatory cytokines. However, they are actually anti-inflammatory cytokines. The authors should confirm and make corrections.

(3)  In the first paragraph of page 4, the authors mentioned that “Targeting these various pathways, including by blockade of PD-1, PD-L1, and CTLA-4, has led to the development of ICIs”. This description seems not clear. Do the authors mean that the ICI treatment is developed by blockade of PD-1, PD-L1 and CTLA-4, or combination of the immune checkpoint inhibitors with the aforementioned signaling pathway inhibitors?

(4)  In table 2, the fourth trial, in the column of agents, the name of agents should be provided, if available.

(5)  In the first column of table 3, the citation information should be indicated as in the reference column of table 1

Author Response

  1. Page number and line number should be provided.

Response: these have been added

  1. In the first and fifth paragraphs of page 3, the authors described that the IL-10 and TGF-b are pro-inflammatory cytokines. However, they are actually anti-inflammatory cytokines. The authors should confirm and make corrections.

Response: the wording of this sentence (lines 100-103) has been adjusted to clarify and confirm the above. In paragraph 5 (line 135) “pro-inflammatory” has been changed to “anti-inflammatory.”

  1. In the first paragraph of page 4, the authors mentioned that “Targeting these various pathways, including by blockade of PD-1, PD-L1, and CTLA-4, has led to the development of ICIs”. This description seems not clear. Do the authors mean that the ICI treatment is developed by blockade of PD-1, PD-L1 and CTLA-4, or combination of the immune checkpoint inhibitors with the aforementioned signaling pathway inhibitors?

Response: this has been edited for clarity

  1. In table 2, the fourth trial, in the column of agents, the name of agents should be provided, if available.

Response: per the trial description at https://clinicaltrials.gov/ct2/show/NCT03817736 - the ICI is not specified. This has been noted in the table.

  1. In the first column of table 3, the citation information should be indicated as in the reference column of table 1

Response: citations added (per MDPI style). Unable to add as superscript, this may have to be manually adjusted, like for Table 1.

Reviewer 3 Report

It is a well-written comprehensive review paper that discusses the role of immunotherapy in the setting of liver transplantation.

1. Please use more recent study, HCC is the 3rd leading (not 4th) cause of cancer related death according to global cancer 2020 statistics (GLOBOCAN)

2. How is the efficacy of combination therapy of ICI different from monotherapy of ICI in the setting of liver transplantation?

3. How is the efficacy of combination of ICI with TKI such as sorafenib, Lenvatinib etc for liver transplantation? Please include the following clinical trial (ClinicalTrials.gov ID NCT03299946) that studies cabozantinib with nivolumab as neoadjuvant

4. Please include what is the general timeline of the recurrence of HCC in patients?

5. What is the correlation of the time when the ICI was started post LT to that of efficacy?

6. Quantitative systems pharmacology models (QSP) are in actively being used in both pharma industry and academic research. Recently the first ever QSP model for HCC was published (PMID: 36323435) that predicts the outcome the clinical trial. I strongly recommend that authors can propose that this HCC QSP models combined with liver resection models (PMID: 24446196, PMID: 31131255, PMID: 34894786, PMID: 30651095) can assist predict optimal treatment regimen, duration for immunotherapy etc and predict the efficacy/treatment plan following liver resection.

Author Response

  1. Please use more recent study, HCC is the 3rd leading (not 4th) cause of cancer related death according to global cancer 2020 statistics (GLOBOCAN)

Response: This has been edited. Ref 1 has been added and all subsequent references have been renumbered (including for Table 1 – manually).

  1. How is the efficacy of combination therapy of ICI different from monotherapy of ICI in the setting of liver transplantation?

Response: given the limitations of the data, in the form of case reports and case series, it is not possible to compare specific regimens in the pre- and post-LT setting. This has been noted in the review now (lines 322-324).

  1. How is the efficacy of combination of ICI with TKI such as sorafenib, Lenvatinib etc for liver transplantation? Please include the following clinical trial (ClinicalTrials.gov ID NCT03299946) that studies cabozantinib with nivolumab as neoadjuvant

Response: this trial has been included in the discussion in section 4, lines 257-263. Table 2 details the upcoming studies (not yet completed or published) that will answer this question.

  1. Please include what is the general timeline of the recurrence of HCC in patients?

Response: this is described in the first paragraph of section 5, lines 353-357.

  1. What is the correlation of the time when the ICI was started post LT to that of efficacy?

Response: as described in section 5, given the limitations of the data (case reports and case series), it is not possible to definitively correlate timing of ICI initiation to its efficacy in treatment of recurrent HCC.  As a corollary, it does appear that early recurrences treated with ICIs suffered more episodes of rejection, and higher associated mortality.

  1. Quantitative systems pharmacology models (QSP) are actively being used in both pharma industry and academic research. Recently the first ever QSP model for HCC was published (PMID: 36323435) that predicts the outcome the clinical trial. I strongly recommend that authors can propose that this HCC QSP models combined with liver resection models (PMID: 24446196, PMID: 31131255, PMID: 34894786, PMID: 30651095) can assist predict optimal treatment regimen, duration for immunotherapy etc and predict the efficacy/treatment plan following liver resection.

Response 6: QSP modeling has been added to the future directions section as a possible complement to traditional trials (lines 425-429).